# A Comparative Evaluation of Techniques for Locating Out-of-View Targets in Virtual Reality

Sathaporn "Hubert" Hu*
Dalhousie University

Joseph Malloch†
Dalhousie University

Derek Reilly‡
Dalhousie University

## ABSTRACT

In this work, we present the design and comparative evaluation of techniques for increasing awareness of out-of-view targets in virtual reality. We first compare two variants of SOUS–a technique that guides the user to out-of-view targets using circle cues in their peripheral vision–with the existing FlyingARrow technique, in which arrows fly from the user's central (foveal) vision toward the target. fSOUS, a variant with low visual salience, performed well in a simple environment but not in visually complex environments, while bSOUS, a visually salient variant, yielded faster target selection than both fSous and FlyingARrow across all environments. We then compare hybrid techniques in which aspects of SOUS relating to unobtrusiveness and visual persistence were reflected in design modifications made to FlyingARrow. Increasing persistence by adding trails to arrows improved performance but there were concerns about obtrusiveness, while other modifications yielded slower and less accurate target acquisition. Nevertheless, since fSOUS and bSOUS are exclusively for head-mounted display with wide field-of-view, FlyingARrow with trail can still be beneficial for devices with limited field-of-view.

**Index Terms:** Human-centered computing—Virtual Reality

## 1 INTRODUCTION

Locating and selecting out-of-view targets without prior knowledge of their positions is a demanding task–particularly in a virtual reality (VR) environment displayed on a commodity head-mounted display (HMD) with a limited field of view (FoV) [7]. However, we can also use VR to augment our field of view with artificial visual cues that assist us in finding and select the target (e.g., [7, 8, 30]). Building on prior work employing visual effects in peripheral vision to enhance awareness of off-screen objects or action ( [19, 29]), and taking advantage of the recent emergence of commodity HMDs with wide FoV, we designed Sign-of-the-UnSeen (SOUS) to allow a user to become aware of and then acquire of out-of-view objects. The cue strictly resides within the user's peripheral vision and moves radially around the user's gaze to inform the user of the position of an out-of-view target. To explore the extent to which SOUS can be unobtrusive to the VR scene while remaining useful we compare two variants: bold SOUS (bSOUS), an opaque circular cue, and faint SOUS (fSOUS), a transparent circular cue. Figure 1 contains the screenshots of the techniques in action.

We conducted two experiments. In the first experiment, we compared bSOUS and fSOUS with a modified version of FlyingARrow (FA) [7]. FA consists of a 3D arrow that flies toward a target. In the experiment participants selected targets placed in their central vision, and periodically acquired off-screen targets indicated by one of the three techniques. Participants acquired off-screen targets faster

---

*e-mail: hs.hu@dal.ca

†e-mail: jmalloch@dal.ca

‡e-mail: reilly@cs.dal.ca

using bSOUS and fSOUS than with FA, as many participants would visually track the arrow to its destination. While the increase in performance is small, it is significant in contexts such as competitive gaming where any performance increase can lead to a victory. Participants sometimes did not notice target cueing with fSOUS in visually complex environments, while bSOUS and FA were more robust to varied detail and changes in the environment. Despite FA being the slowest technique, it had higher overall subjective ratings. While bSOUS compared well to FA in these ratings, many participants found fSOUS frustrating to use.

By being placed in the user's peripheral vision, SOUS is less obtrusive than FA, as it does not block the scene in the user's central vision. In addition, while FA arrows move through the scene in the 3D environment, SOUS cues are presented on an invisible layer in front of the scene (as with a Heads Up Display or HUD): this may further help in distinguishing SOUS as a cue, rather than another object in the scene. SOUS is also more visually persistent than FA, whose arrow flies outside the user's FoV if not followed. If it was possible to provide these qualities to FA, this could be beneficial for HMDs with limited FoVs unable to display a technique that requires the use of far-peripheral vision like SOUS. We therefore extended FlyingARrow (FA) [8] with two behaviours: +Arc, and +Trail. +Arc attempts to make FA less obstructive by making the cue orbit around the user at a set distance–keeping it out of the way of on-screen targets and making it more distinctive as a cue to the user vs. an object in the scene. +Trail makes FA emit a trail, making it more visually persistent–if the user loses sight of the arrow, the trail will linger, allowing the user to remain aware of the target's direction.

In a second experiment, we compared FA, FA+Arc, FA+Trail, and FA+Arc+Trail. This experiment involved the same combination of selecting on-screen and off-screen targets. We found that +Arc slowed down participants and did not make the technique more comfortable to use. While participants found +Arc to be slightly less obtrusive, it was also less robust to visual complexity, suggesting that the intended increase in visual distinctiveness was not achieved. Despite participants stating that +Trail was more obtrusive, including a trail improved the speed of target acquisition.

In the remainder of this paper we discuss related work in visual perception and techniques for off-screen target awareness and acquisition, detail the SOUS and FA designs, describe our experiments and present results. After this we discuss the implications of our findings for the design of cues for off-screen objects in VR.

## 2 RELATED WORK

### 2.1 Visual Perception

Prior literature [15, 20] suggests that the shape of cues in peripheral vision should be simple, because it is difficult to distinguish complex shapes in this region. Luyten et al. [15] performed an experiment where each participant wore a pair of glasses. On each side of the glasses, there was a colour screen that could display a shape. Each shape was positioned almost 90°away from the foveal center of vision. They found the participants could recognize that the shapes were different but had difficulty recognizing composite shapes. While our ability to distinguish shapes is reduced in peripheral vision [20], it is adapted to notice sudden changes [12]. Work by Bartram et al. [2] suggests animation can further enhance

awareness of new objects in the peripheral vision, while Luyten et al. [15] found that blinking is effective in the peripheral region to notify the user of changes.

Ball and North [1] conducted a study to investigate why users have better target acquisition performance on larger screens, finding that the improved performance was due to peripheral awareness of content that they could rapidly focus on. Although bSOUS and fSOUS are VR techniques, they build on Ball and North's observations by making use of the user's peripheral vision and providing support for rapidly locating the indicated off-screen target.

Visual cues can interact with other objects in the visual field, impacting their ability to capture attention. According to Rosenholtz et al. [21], whether we will find a target or not depends on the *visual salience* of the environment. Salience indicates how different the target is from its environment. For example, a low-salience target tends to have a similar visual appearance to or blend in with the environment. Experiments by Neider and Zelinsky [18] support this by demonstrating that it is more difficult for a person to find a target if the background resembles the target. Additionally, Rosenholtz et al. [21], and Rosenholtz [20] present mathematical formulae to quantify visual salience, but they are designed for static images projected onto 2D space and in scenarios without animation. Therefore, they don't directly translate to dynamic VR environments and are not used in our study. More recent work has explored machine learning methods to model visual salience (e.g. [14]); while such techniques are promising ways to measure salience in a given scene, no standard has been established for using these methods to generate scenes with desirable salience attributes in controlled experiments.

Cues with high visual salience tend to be more effective, but such cues are is not always appropriate. For example, in cinematic viewing, users may prefer subtler cues to avoid obstruction and distraction. McNamara et al. [16] designed a study where a part of a 2D screen would be subtly modulated in order to guide the participants toward a certain area. Their study showed some efficacy in modulation. Later on, Rothe and Hußmann [22, 23] conducted an experiment where they used spotlights to subtly guide the user to a target and found them to be effective. We created fSOUS as a subtler way to guide the user to the out-of-view targets. However, unlike the cues explored in these prior studies [16, 22, 23], our cue strictly resides in the user's peripheral vision.

### 2.2 Existing Techniques

Some existing techniques for guiding the user to out-of-view targets, such as EyeSee360 [7], 3D Wedge [30], and Mirror Ball [4], have roots in an earlier technique called Halo [3]. Halo provides cues for off-screen targets on small-screen handheld devices. Halo uses circles to represent the targets, with sections of the circles rendered on the edge of the device. The position and size of a circle indicates a target's position relative to the area displayed on the device. Halo was compared with a technique called Arrow which uses arrows pointing toward the targets labelled with the target's distance. Halo was better at indicating both position and distance in their tests. Burigat et al. [6] compared Halo to a variant of Arrow in which the length of an arrow indicated distance. This allowed participants to more easily rank the distances of the targets but fared worse than Halo for indicating the actual target distances. Schinke et al. [24] developed an handheld Augmented Reality (AR) version of Arrow where 3D arrows point toward AR targets located some distance from the viewer (and often off-screen). The user then uses the device to guide themselves toward the targets. Their evaluation showed the technique to work better than Radar, a technique that provides a simplified overhead view of the area.

Gruenefeld et al. [7] notes that many AR HMDs such as Microsoft HoloLens v1 suffer from limited screen real estate (much like handhelds) and limited FoV. They introduced EyeSee360, an overview technique that allows the user to see out-of-view targets by

representing them as dots on a grid. The dot's position on the grid indicates the target's orientation and distance relative to the viewer. EyeSee360 is a visually obtrusive technique. As such, Gruenefeld et al. [9] suggest that the user should be able to set the cue's visibility on an "on-demand" basis. They compared EyeSee360 against Halo, Wedge (a variant of Halo that uses acute isosceles triangles [10]), and Arrow and found it to be the best-performing technique. Greuenefeld et al. [8] later developed FlyingARrow (FA), an animated variant of Arrow, which they found to be slower than EyeSee360. Other overview techniques explored in the literature include Radar and 3D Radar [9], and Mirror Ball [4], which presents a distorted view of the surroundings rendered as a ball.

Yu et al. [30] proposed a 3D variant of Wedge for use in VR, to indicate relative position and distance of targets. Unlike the original Wedge, the cue for 3D Wedge appears in front of the user instead of around the edges of the screen. Each wedge is also a 3D pyramid whose base is pointing the toward the target, and whose size indicates the distance. The researchers found that 3D Wedge was more effective at finding targets than overview techniques such as Radar–except when there were many targets. They improved 3D Wedge by embedding an arrow pointing toward the target inside each pyramid.

Unlike the techniques covered so far which focus on target acquisition, Xiao and Benko [29], and Qian et al. [19] implemented techniques for increasing the user's awareness of off-screen objects without requiring the user to select them. Xiao and Benko [29] added a small LED light grid around the main HMD display. Although the grid had low resolution, it was sufficient for the user to glean additional information in their peripheral vision based on colour and lighting changes. Qian et al. [19] used a similar approach for object awareness specifically: when there is an object close to the user, a corresponding area of the screen's edges lights up. Their evaluation found that this allowed users to notice off-screen targets.

## 3 TECHNIQUES

### 3.1 bSOUS and fSOUS

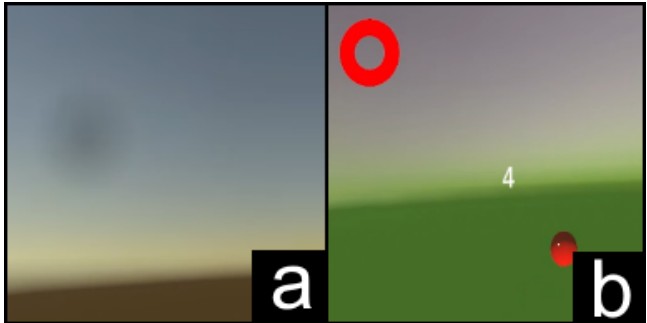

Figure 1: a: fSOUS. b: bSOUS.

Sign of the UnSeen (SOUS) is a family of peripheral vision-based techniques that include bSOUS and fSOUS. When a target of interest appears off-screen, a SOUS cue appears in the user's peripheral vision. The cue moves radially based on the user's relative position to the target. For example, if the target is slightly above and left of the user, the cue will appear on the left side and will be rotated slightly upward around the user's forward gaze cursor. Although we would like the cue to be as far away as possible from the user's foveal vision, there is currently no commercially-available VR headset that encompasses the full human visual field (about 105° from the center of foveal vision [26]). Nevertheless, some commercial headsets (e.g., the Pimax 5K Plus used in this study) can display what is considered to be in the far-peripheral. As such, the SOUS cue is

located around 60°from the center which is considered to be in peripheral vision [26] and is displayable by commercially available VR headsets.

fSOUS is semi-transparent and subtle, and intended to support scenarios such as cinematic viewing in which more explicit cues might be highly disruptive to the viewing experience. We conducted a small pilot experiment with 5 participants to determine a lower bound opacity level that was still detectable. All five participants found that they could see the cue at 5% opacity within a minimal skybox environment(Figure 4:a). While we also found that closer to 50% opacity would be readily detectable in a more complex environment like Mixed (Figure 4:d), we maintain the 5% level in our experiment across environments. The circular cue uses a radial gradient that shifts between black and white at 5.56 Hz.

bSOUS appears as a ring that blinks from red to white at 1.11 Hz. The cue is opaque and blinks rather than gradually changing colour, making the cue more immediately noticeable in peripheral vision. bSOUS uses a ring instead of the solid circle used by fSOUS because without transparency a solid circle is not visually distinct enough from the targets used in our experiments.

## 3.2 FlyingARrow (FA)

FA is a further refinement of Arrow [24] for immersive AR explored by Gruenefeld et al [8]. FA's cue is a 3D arrow that flies toward the target, using animation to encourage the user to act. The cue would play a sound once it collides with the target and then subsequently disappeared.

FA was designed for AR devices with small FoVs like Microsoft Hololens v1, and we adapt FA for HMDs with larger FoVs. In the original version FA arrows start in a corner of the screen and move across the user's limited screen space, to allow the user time to perceive and interpret the cue. Given the increased FoV, in our variant the arrow starts 1m in front of the user (in virtual space) in the center of the screen. We also removed the sound effect as it was a potential confounding factor: all off-screen targets were equidistant from the user in our experiments, eliminating the need for a distance indicator (the role of the sound). We reused the 3D arrow asset used by Gruenefeld et al. [8], available in: `https://github.com/UweGruenefeld/OutOfView`. The 3D arrow can be seen in Figure 2. While the red colour of the model could introduce a colour confound, we decided to not change the colour so the cue would have the same appearance with the original implementation.

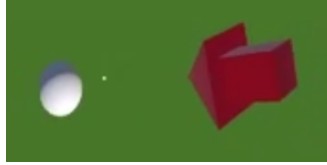

Figure 2: FlyingARrow as appeared in the first part of the study. The arrow is travelling toward the target.

We now describe the +Arc and +Trail modifications to FA that we explore in the second experiment. As discussed, +Arc was designed to make FA less obstructive to and more visually distinguished from on-screen targets by orbiting around the user toward the target. While the standard FA cue starts 1 m in front of the user and travels straight to the target at the speed of 10 m/s, a +Arc cue starts at $x$ metres away from the user, where $x$ is the physical distance from the user to the target. In our experiment, this is 5m for all off-screen targets, placing the cue behind the on-screen targets. The cue's physical size is then adjusted to make sure that it has the angular size of 5°, equal to the cue's default angular size at 1m. The cue then orbits around the user at the speed of at $tan^{-1}\left(\frac{10m/s}{x}\right)$ around

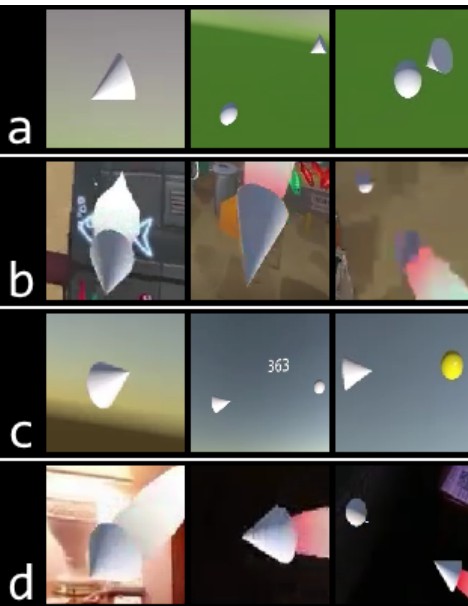

Figure 3: The variations of FA cues at the various stages before reaching the target. a: FA-Arc-Trail in Training environment. b: FA-Arc+Trail in Mixed environment. c: FA+Arc-Trail in None environment. d:FA+Arc+Trail in Hotel environment. For more information about the environment and the credits to the graphical assets, please refer to Section 4.

the user's upward vector. This vector is recalculated at every frame update.

+Trail makes the FA cue's visibility persist longer by emitting a trail. The standard FA cue is not visible to the user once it leaves the screen. The trail has the following Unity properties: Widest Size = 0.315m, Narrowest Size = 0.315m, Time = 5s. This trail allows the user to maintain awareness of an FA cue and to follow the trail to relocate it. We altered the shape of the cue from an arrow to a cone in our second experiment across all conditions to improve the visual integration of the trail and main cue. We changed the colour from red to white to eliminate the colour confound for the second experiment.

Below is the summary of four variations of FA based on the new behaviours:

- FA-Arc-Trail: The cone travels straight and directly to the target without any trail, similar to FA used in the first study.

- FA+Arc-Trail: The cone orbits around the user (rotating around user's upward vector $\vec{u}$ at the moment the target first appears) to reach the target. It does not leave a trail.

- FA-Arc+Trail: The cone travels straight to the target, leaving a trail.

- FA+Arc+Trail: The cone orbits around the user to reach the target (rotating around user's upward vector $\vec{u}$ at the moment when the target first appears) and leaves behind a trail.

## 4 ENVIRONMENTS

Prior work [18] suggests that the visual complexity of the environment may impact user performance. In order to explore how techniques interact with environment complexity we varied environment as an experimental factor in our study. We created three types of environment: None, Hotel, and Mixed. The details of the environments are as follows:

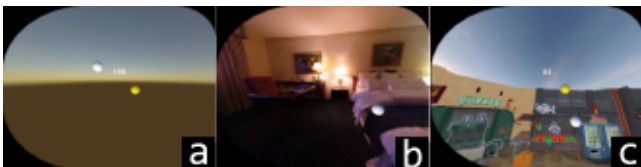

Figure 4: Screenshots of the environments as they appeared in our implementation (the left eye/screen is shown here). a: None. b: Hotel. c: Mixed.

- **None** (Figure 4:a): a generic skybox with a brown horizon and a blue clear sky, representing environments with low visual complexity. Constructed using the default skybox in Unity 2018.3.7f1.

- **Hotel** (Figure 4:b): a photorealistic skybox of a hotel room, representing environments with moderate visual complexity. This skybox is CC Emil Persson (`https://opengameart.org/content/indoors-skyboxes`).

- **Mixed** (Figure 4:c): a combination of a photorealistic skybox and 3D models, some of which are animated; this represents environments with high visual complexity. The 3D models are taken from the Pupil Unity 3D Plug-in (`https://github.com/pupil-labs/hmd-eyes`) and the skybox is CC Emil Persson (`https://opengameart.org/content/winter-skyboxes`).

While each environment differs in visual complexity, we are unable to quantify this precisely, as discussed previously. Instead, including these environments allows us to generally explore the robustness of each technique to typical environmental differences.

## 5 EXPERIMENT 1: COMPARING bSOUS, fSOUS AND FLYINGARROW

We performed the first experiment to evaluate our techniques bSOUS and fSOUS against an existing technique called FA. Furthermore, since bSOUS and fSOUS have different visual salience (achieved through differences in animation and opacity), we explore the impact of a peripheral cue's visual salience on target acquisition.

### 5.1 Research Questions

**RQ1.1: How do the techniques affect target acquisition performance and the user's cognitive load?** To measure performance, we collect (1) number of successful out-of-view target acquisitions, and (2) time to acquire an out-of-view target. We administer the NASA TLX to assess cognitive load. An ideal cue has fast acquisition times, high success rate, and low cognitive load.

**RQ1.2: How do the techniques interact with the visual scene?** We measure how the environments affect (1) number of successful out-of-view target acquisitions, and (2) time to acquire an out-of-view target. An ideal cue works well under a range of visual scenes.

**RQ.13: What are the subjective impressions of the cues?** We gather subjective feedback through questionnaire and interview. An ideal cue provides a positive experience for the user. A cue with good performance may be less viable than a technique with inferior performance that is preferred by the user.

### 5.2 Participants

We conducted the first study at a research university with 24 participants. We recruited the participants using an email list for graduate students in the faculty of computer science at our institution. Six participants were female, and 14 were males. Four participants did not indicate their gender. 19 participants indicated that they had a prior experience using a VR headset, and seven indicated that

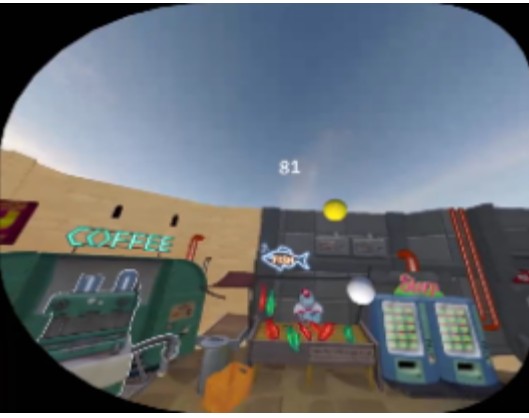

Figure 5: A screenshot of the interface taken from the left-side screen of P15. The current environment of the screen is Mixed. The number shows the current score that the participant currently had. The spheres represent the in-view targets. One of the target was yellow, because the participant was dwelling on it. The credits to the skybox photo and the 3D assets are available in Section 4.

they had participated in a VR study before. The median score for self-reported VR proficiency level is 4 out of 7.

### 5.3 Software and Hardware Instrument

We used the Pimax 5K Plus for the study because it has a wider FoV than most commercially available headsets. Pimax 5K Plus has the diagonal FoV of 200° [27]). The diagonal FoVs of other popular and widely available HMDs are: Occulus Rift DK1 - 110° [5] , HTC Vive - 110° [5], Microsoft HoloLens v1 - 35° [28].

During the training and the trials, each participants interacted with a VR interface implemented using Unity 2018.3.7f1 and SteamVR. The interface had a score on the top-left corner of the screen to keep the participants engaged. The targets were 3D spheres that the participants could select by rotating their head to land the cursor onto the target (gaze cursor). The gaze cursor is a circle 1.2m in front of the user with the size of 0.01m. This means that it has the angular size of 0.57°. Based on the condition, the participant would be operating inside a specific virtual environment and could avail themselves to one of the techniques to select out-of-view targets. We used R to analyze the data collected while the participant was performing the tasks. We also used it to analyze NASA-TLX raw scores and the questionnaire answers.

### 5.4 Questionnaire Instrument

After completing each technique, each participant must complete NASA-TLX Questionnaires (More information in Hart [11]) and 7-point Likert scale questions:

- **S1**: The technique is overall effective for helping me to locate an object.

- **S2**: I can immediately understand what the technique is telling me.

- **S3**: The technique precisely tells me where the target is.

- **S4**: The technique helps me to rapidly locate the target.

- **S5**: The technique makes it difficult to concentrate.

- **S6**: The technique can be startling.

- **S7**: The technique gets my attention immediately.

- **S8**: The technique gets in the way of the virtual scene.

- **S9**: The technique makes me aware of the objects outside the FoV.
- **S10**: The technique is uncomfortable to use.

For each Likert-scale question, each participant would rate the statement from 1 to 7 with 1 being "completely disagree" and 7 being "completely agree."

## 5.5 Procedure

### 5.5.1 Overview

The steps were as follows: **STEP 1 –** The participant provided informed consent and completed the background questionnaire. **STEP 2 –** Then, we trained a participant to use one of the three techniques (bSOUS, fSOUS, FA) by asking them to select 10 out-of-view targets in the training environment while trying simultaneously to select as many in-view targets as possible. During the training, we primed the participant to prioritize selecting out-of-view targets. If the participant failed to become familiar with the technique, they would repeat the training trials. **STEP 3 –** The participants would complete the actual trials by selecting 20 out-of-view targets while trying to simultaneously select as many in-view targets as possible in one of the three environments (None, Hotel, Mixed). After the 20 trials were completed, we altered the environments. We repeated these steps until the participant experienced all of the environment with the technique. The next section, Section 5.5.2, contains additional information on how a participant would complete a trial during the study. **STEP 4 –** Afterward, the participant completed a NASA-TLX instrument and the 10 Likert scale question. We allowed the participants to take a short break of several minutes before proceeding. **STEP 5 –** The participant then repeated **STEP 2** to **STEP 4** until they experienced all the techniques.

Since there were 20 trials for each environment and each technique, a participant would have completed $3 \times 3 \times 20 = 180$ trials. We used Latin squares to arrange the ordering of the techniques and the environments which resulted in nine orderings of the conditions.

### 5.5.2 Completing a Trial

The main task of the study involved target selection. Each participant selected a target via gaze selection by dwelling a cursor onto the target for 500 milliseconds. There were two types of selection in the study: in-view and out-of-view targets. We considered a selection of an out-of-view targets as a trial for our studies. While we asked our participants to select both types of targets, we also primed our participants to prioritize selecting out-of-view targets. We also told the participants that they would earn more points by selecting out-of-view targets, and the targets could disappear before a successful selection. We made the targets disappear to encourage the participants to find the targets as quickly as possible.

The in-view targets spawned in front of the participant (within 40° of the user's forward vector) every one to two seconds. They

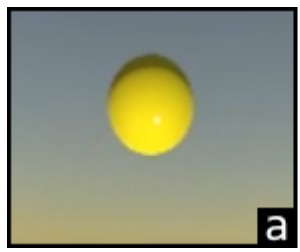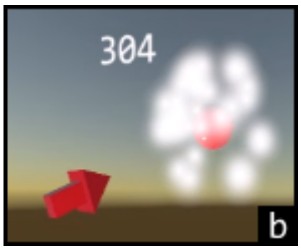

Figure 6: a: When the head cursor lands on the target, the target turns yellow. The participant must dwell for 500 milliseconds to select it. b: Upon a successful selection, the target turns red. An in-view target simply fades after turning red while an out-of-view also sparkles.

had one second to select the target before it would disappear. These targets could appear in any direction. An in-view target was worth one point. The out-of-view targets spawned at least 80° away from the participants' forward vectors in any direction. Since the targets were further away, the participants must use a technique to locate the targets. The spawning rate for this type of target was every 5.5 to 6.5 seconds. The participant had five seconds to select an out-of-view target after it spawned. Since the out-of-view targets were further away in term of angular distance, longer time was required. This type of target was worth 10 points. Both types of targets have the same appearance before selection which are white spheres with the angular size of 7°(The physical sizes of the spheres continuously rescale to maintain the angular size). The pilot study determined that the size was reasonable for the tasks. The only visual difference between an in-view and an out-of-view target was that the in-view target faded upon selection whereas the out-of-view target sparkled (Figure 6:b). We decided to make the target appearances the same, because we were controlling for visual salience.

We used the out-of-view target selection task to evaluate the performance and efficacy of the techniques. The in-view targets encouraged participants to return to the original orientation, and dissuade the participants from waiting for the next out-of-view target to appear. In our study, we considered an attempt to select an out-of-view target a trial. We considered a trial to be successful if the participant could dwell on the target long enough that it would trigger the selection animation. We considered a trial to be unsuccessful if the participant could not dwell on the target long enough to trigger the animation or if they could not locate the target. A trial completion time for a successful target selection was the duration from when the target first spawned and until when the participant landed the cursor onto the target. This excludes the dwell time and the animation time.

## 5.6 Results

### 5.6.1 Number of Unsuccessful Target Acquisition

To answer **RQ1.1–1.2**, we recorded the numbers of failed out-of-view target acquisition or the numbers of failed trials per participant. For the number of unsuccessful target selection, we used `lme4` to model a mixed logistic regression that predicted the probability of failure with the following factors: the techniques, the environments, with the participants as the random effect. Then, we computed pseudo-$r^2$ for the model using `MuMIn` which implements an algorithm found in Nakagawa, Johnson and Schietzeth [17].

The model that we obtained after the fitting is as following: $\text{logit}(\Pr(Fail)) = -3.85 + 0.43 \times fSOUS + 0 \times bSOUS + 0.06 \times Hotel - 0.13 \times Mixed + 7.65 \times fSOUS:Hotel + 1.38 \times bSOUS:Hotel + 6.75 \times fSOUS:Mixed + 1.22 \times bSOUS:Mixed$. The coef-

| | $\beta$ | OR | SE | Z | p |
|---|---|---|---|---|---|
| (Intercept) | -3.85 | 0.02 | 0.34 | -11.21 | 0.00* |
| **Techniques** | | | | | |
| fSOUS | 0.43 | 1.53 | 0.33 | 1.31 | 0.19 |
| bSOUS | 0.00 | 1.00 | 0.35 | 0.00 | 1.00 |
| **Environments** | | | | | |
| Hotel | 0.06 | 1.06 | 0.35 | 0.18 | 0.86 |
| Mixed | -0.13 | 0.88 | 0.36 | -0.37 | 0.71 |
| **Tech. : Env.** | | | | | |
| fSOUS:Hotel | 7.65 | 2.04 | 0.43 | 4.78 | 0.00* |
| bSOUS:Hotel | 1.38 | 0.32 | 0.48 | 0.67 | 0.50 |
| fSOUS:Mixed | 6.75 | 1.91 | 0.44 | 4.34 | 0.00* |
| bSOUS:Mixed | 1.22 | 0.20 | 0.50 | 0.39 | 0.70 |

Table 1: The summary of the coefficients, ORs, and other information for fitting a mixed multiple linear that predict the probability of failing to acquire an out-of-view targets based on the techniques and the environments. * signifies that $p \leq 0.05$.

ficients, odd ratios (OR), standard errors (SE), and other information are summarized in Table 1. The $r^2$ are as following: theoretical marginal = 0.06, theoretical conditional = 0.39, delta marginal = 0.06, delta conditional = 0.14. The most important effect size for interpretation is the theoretical conditional $r^2$ since it represents the variance explained by the entire model including the random effect. Since it is 0.39, it indicates that the techniques and the visual scenes had moderate effect on the success of target selection. However, the theoretical marginal $r^2$, or $r^2$ that excludes the random effect of the participants is only 0.06–meaning that there is a strong effect from each individual themselves.

Based on Table 1, we found that there was a strong interaction between the environments and fSOUS. The participants failed more often while using fSOUS with Hotel and Mixed. Since the participants ($n = 14$) indicated during the interviews that they often found fSOUS cues blending into the environment, we conclude that the faint nature of the cue led the participants to lose sight of the cue and subsequently failed to select the targets.

Regarding **RQ1.1** we observed that target acquisition success differed between techniques, but this was also conditioned on the visual scene or the environment which ties directly to **RQ1.2**. We found that the visual scene could affect how the participants perceived the cue and subsequent success in target acquisition. We found that despite bSOUS and fSOUS have very similar cueing mechanism, they have very different performance in term of target acquisition success in different environments.

### 5.6.2 Time for Target Acquisition

In addition to the probability of target acquisition, we also considered the time of target acquisition as another important measure to answer **RQ1.1–1.2**. We measured the time the participants took to reach the targets in successful trials (ie. excluding the 500ms dwelling time). Then, we fitted a mixed multiple linear regression model using the participants as the random effect. We studied the following variables: (1) the techniques, (2) the environments, and (3) the angular distance between the user's initial position to the target. Even though our main focuses are the techniques and the environments, we also have to discuss distance as the out-of-view targets had different distances from the participants. We did not have to consider Fitts's Law for this study, because our targets have the same angular size.

We did not normalize the data because of the suggestion made by Schmidt and Finan [25]. They argue that if the sample size is sufficiently large, normalization could introduce a statistical bias when fitting a linear model. The model that we fitted using `lme4` was as following: $Time = 2.19 - 0.14 \times bSOUS + 0.31 \times fSOUS - 0.06 \times None - 0.03 \times Mixed + 0.01 Dist + 0.09 \times bSOUS : None - 0.34 \times$

| | $\beta$ | SE | df | t | p |
|---|---|---|---|---|---|
| (Intercept) | 2.19 | 0.10 | 57.73 | 22.58 | 0.00* |
| **Techniques** | | | | | |
| bSOUS | -0.14 | 0.05 | 3927.89 | -2.81 | 0.00* |
| fSOUS | 0.31 | 0.05 | 3928.12 | 5.94 | 0.00* |
| **Environments** | | | | | |
| None | -0.06 | 0.05 | 3927.91 | -1.30 | 0.19 |
| Mixed | -0.03 | 0.05 | 3927.91 | -0.66 | 0.51 |
| **Angular Distance** | | | | | |
| Dist | 0.01 | 0.00 | 3930.34 | 13.62 | 0.00* |
| **Techn. : Env.** | | | | | |
| bSOUS:None | 0.09 | 0.07 | 3927.88 | 1.36 | 0.17 |
| fSOUS:None | -0.34 | 0.07 | 3927.99 | -4.73 | 0.00* |
| bSOUS:Mixed | 0.07 | 0.07 | 3927.93 | 0.97 | 0.33 |
| fSOUS:Mixed | 0.02 | 0.07 | 3928.04 | 0.28 | 0.78 |

Table 2: The summary of coefficients and the results of the tests for the coefficients for the time required to reach out-of-view targets. We only considered successful trials for analysis. * signifies $p \leq 0.05$.

$fSOUS : None + 0.07 \times bSOUS : Mixed + 0.02 fSOUS : Mixed$. The $r^2$ of the model computed using `MuMIn` were: marginal = 0.06, conditional = 0.25. The conditional $r^2$ indicated that the model was moderately decent at explaining the time required to reach target. Table 2 show the results of the tests on the coefficients. The coefficient representing angular distance ($\beta = 0.01$, $t(3930.34) = 13.62, p \leq 0.05$) was small when compared to other coefficients. We found that there was an interaction effect between the techniques and the environments. Particularly, fSOUS was faster in None ($\beta = -0.34, t(3927.99) = -4.73, p \leq 0.05$). The participants ($n = 14$) indicated that fSOUS cue blending with more visually complex environments (Mixed, Hotel) caused them to be slower. We found that in term of main effects, the techniques were statistically significant with bSOUS being the fastest, FA being the second fastest, and fSOUS being the slowest. On the other hand, the main effects from the environment are not statistically significant.

Interestingly, some participants indicated during the interviews FA gave them more time to reach the target ($n = 4$) when this was actually not the case. Some ($n = 4$) felt that the speed of the cue influenced their own target acquisition speed–making them slower.

### 5.6.3 Cognitive Load

To further answer **RQ1.1**, we collected and analyzed the participants' cognitive load after using a technique using NASA-TLX scores. The median NASA-TLX raw scores are as follows: FA = 42, fSOUS = 70.5, bSOUS = 57.5. This suggests that fSOUS and bSOUS induced higher cognitive load than FA. ART ANOVA with repeated measure (using `art`) revealed significant differences between the techniques ($F(2, 18.63) = 7.40, p \leq 0.05$). Pairwise comparisons with Tukey adjustment (using `emmeans`) showed that there were significant differences between FA and fSOUS ($c = -21.0, t_{ratio}(46) = -6.0, p \leq 0.05$), and fSOUS and bSOUS ($c = 13.9, t_{ratio}(46) = 3.92, p \leq 0.05$). The difference between FA and bSOUS was not statistically significant ($c = -7.1, t_{ratio}(46) = -2.03$). The interview data from 16 participants suggested that fSOUS induced more cognitive load, because the cue tended to blend with the environment which forced them to simultaneously find the cue and the target.

### 5.6.4 Questionnaire

To answer **RQ1.3**, we administered a questionnaire after a participant finished using a technique.

**S1:** Many of the participants indicated that FA ($Mdn = 7$) was overall effective at helping. fSOUS was considerably less effective ($Mdn = 4$); however, Figure 10 shows that the Likert scores were distributed quite evenly. This indicated that there are mixed opinions from the participants. bSOUS ($Mdn = 6$) was overall more effective than fSOUS, but slightly more less effective than FA.

**S2:** Many of the participants found FA and bSOUS to be very comprehensible ($Mdn$: FA = 7, bSOUS = 6). Interestingly, on average, they found fSOUS to be less comprehensible ($Mdn = 4.5$) than bSOUS despite that it uses the same mechanism to provide the location information of the out-of-view targets. The scores for fSOUS were somewhat evenly distributed (Figure 10) which indicates mixed opinions among the participants for fSOUS.

**S3:** Many of the participants found FA and bSOUS to be very precise ($Mdn$: FA = 7, bSOUS = 6). Interestingly, the participants overall found fSOUS ($Mdn = 4.5$) to be less precise despite that it had the same cueing mechanism with fSOUS.

**S4:** Many of the particants found FA and bSOUS were helping them to quickly acquire the out-of-view targets ($Mdn$: FA = 7, bSOUS = 6). fSOUS was less effective ($Mdn = 3$) despite it had the same cueing mechanism with bSOUS. However, some participants still found fSOUS to be effective.

**S5:** The medians ($Mdn$: FA = 3.5, fSOUS = 3, bSOUS = 2) indicated that, overall, the three techniques did not negatively affect

| Q | Techniques | 1 | 2 | 3 | 4 | 5 | 6 | 7 |
|---|---|---|---|---|---|---|---|---|
| 1 | FA | 0 | 0 | 0 | 2 | 1 | 3 | 18 |
|  | fSOUS | 2 | 5 | 2 | 5 | 6 | 1 | 3 |
|  | bSOUS | 0 | 0 | 0 | 1 | 8 | 5 | 10 |
| 2 | FA | 0 | 0 | 1 | 0 | 1 | 3 | 19 |
|  | fSOUS | 4 | 3 | 1 | 4 | 5 | 5 | 2 |
|  | bSOUS | 0 | 1 | 0 | 2 | 5 | 7 | 9 |
| 3 | FA | 0 | 0 | 0 | 0 | 3 | 2 | 19 |
|  | fSOUS | 3 | 6 | 5 | 2 | 3 | 2 | 3 |
|  | bSOUS | 0 | 2 | 2 | 4 | 2 | 5 | 9 |
| 4 | FA | 0 | 1 | 3 | 1 | 2 | 3 | 14 |
|  | fSOUS | 4 | 7 | 2 | 2 | 3 | 3 | 3 |
|  | bSOUS | 0 | 0 | 3 | 0 | 3 | 8 | 10 |
| 5 | FA | 5 | 3 | 4 | 2 | 3 | 6 | 1 |
|  | fSOUS | 5 | 2 | 6 | 3 | 3 | 2 | 3 |
|  | bSOUS | 9 | 6 | 1 | 1 | 3 | 2 | 2 |

| Q | Techniques | 1 | 2 | 3 | 4 | 5 | 6 | 7 |
|---|---|---|---|---|---|---|---|---|
| 6 | FA | 3 | 5 | 1 | 6 | 6 | 1 | 2 |
|  | fSOUS | 5 | 4 | 5 | 5 | 4 | 0 | 1 |
|  | bSOUS | 4 | 6 | 4 | 3 | 3 | 3 | 1 |
| 7 | FA | 0 | 0 | 1 | 0 | 1 | 9 | 13 |
|  | fSOUS | 11 | 2 | 6 | 1 | 1 | 2 | 1 |
|  | bSOUS | 0 | 0 | 1 | 3 | 4 | 7 | 9 |
| 8 | FA | 2 | 3 | 4 | 1 | 4 | 4 | 6 |
|  | fSOUS | 13 | 2 | 4 | 2 | 1 | 2 | 0 |
|  | bSOUS | 8 | 4 | 7 | 2 | 1 | 1 | 1 |
| 9 | FA | 0 | 0 | 0 | 1 | 2 | 5 | 16 |
|  | fSOUS | 2 | 3 | 2 | 5 | 6 | 4 | 2 |
|  | bSOUS | 1 | 0 | 0 | 1 | 4 | 8 | 10 |
| 10 | FA | 10 | 5 | 3 | 2 | 2 | 2 | 0 |
|  | fSOUS | 1 | 4 | 5 | 4 | 1 | 2 | 7 |
|  | bSOUS | 7 | 4 | 6 | 2 | 3 | 2 | 0 |

Figure 7: The heatmap represents the frequencies of the responses for the Likert scale statements. Red means less participants and green means more participants. The numbers indicate indicates the frequencies of responses.

their concentration. However, Figure 10 indicated that bSOUS had the best performance in this regard.

**S6:** The participants indicated that on average (*Mdn*: FA = 4, fSOUS = 3, bSOUS = 3), all techniques were almost equally startling. We found this result to be interesting. We expected fSOUS to be the least startling, because its cue was faint. Figure 10 indicates somewhat even distributions of scores for all three techniques. The median score for FA was somewhat surprising as we expected the participants to find the technique more startling. Because unlike SOUS, FA could travel very close to the user or even through the user. However, during the interviews, only few participants ($n = 3$) indicated this to be an issue.

**S7:** FA ($Mdn = 7$) and bSOUS ($Mdn = 6$) were similarly effective at grabbing the participants' attention whereas fSOUS ($Mdn = 2$) was less effective. It was not surprising for FA to be more attention-grabbing than fSOUS as the FA cue initially appears in the user's foveal vision as opposed to their peripheral vision. However, we found FA and bSOUS's similar effectiveness to be surprising.

**S8:** Most of the participants indicated that FA ($Mdn = 5$) was more obstructive than fSOUS ($Mdn = 1$) and bSOUS ($Mdn = 2.5$). This result means that peripheral-based techniques are beneficial at reducing visual obstruction.

**S9:** Most of the participants indicated that FA ($Mdn = 7$) and bSOUS ($Mdn = 6$) were effective at making them aware of out-of-view targets. At a glance, fSOUS ($Mdn = 4.5$) seemed to not make the participants aware of the out-of-view targets. However, Figure 10 indicates a bimodal distribution for fSOUS–meaning that some participants found this technique helping them to become aware of out-of-view targets whereas some did not find it helpful.

**S10:** FA ($Mdn = 2$) and bSOUS ($Mdn = 3$) were similar in term of comfort while fSOUS ($Mdn = 4$) was slightly more uncomfortable to use. We noticed from Figure 10 that some participants found fSOUS to be very uncomfortable to use while some participants found it to be as comfortable to use as the other techniques.

## 6 EXPERIMENT 2: COMPARING THE VARIANTS OF FLYING-GARROW

We performed the second experiment to observe if we could improve FA using certain properties of bSOUS and fSOUS. While we were implementing FA and analyzing its design, we realized that FA has

two issues. First, the cue is not visually persistent–if the user loses sight of it, they may not be able to relocate it. The original implementation uses a sound cue to alleviate this issue. However, since our study was purely about visual cue, we decided to implement +Trail behaviour instead. The second issue is that the cue can travel too close or even cut through the user which led us to implement +Arc behaviour.

We noted bSOUS and fSOUS have visually persistent cue and the position of the cue are also relative to the user. We compared the four variations of FA: FA-Arc-Trail, FA-Arc+Trail, FA+Arc-Trail, and FA+Arc+Trail in this part of the study. Whereas the FA cue in the first part is an arrow (Figure 2), the FA cue in the second part is a cone (Figure 3) to make its appearance more compatible with a trail. The cue is white so that it would not act as a colour confound.

Experiment 2 was largely similar to the first one. Each participant used a technique in the three environments, completed a NASA-TLX questionnaire and 10 Likert-scale questions, completed an interview, and moved onto the next technique. After completing the first part of the study, each participant would proceed directly to this one after a short break of several minutes (However, each participant could extend the break if they needed more time.). We asked our participants to use the four variations in the three environments to select 20 out-of-view targets per condition. Each participant performed $4 \times 3 \times 20 = 240$ trials in total for the study. Similar to the first study, we also used Latin square to arrange the conditions which resulted in 12 orderings of the conditions. Since the participants were already familiar with the target selection task, we increased the difficulty of the task by decreasing the size of the targets from 7° to 5°. After the completion of this part, each participant received 15 Canadian dollars. The compensation was for their time for both experiments.

### 6.1 Research Questions

**RQ2.1: Do the variations of FA have the same performance and induce the same cognitive load?** If the variations have different performance, they should have different probability of target acquisition failure, and different speed to reach target. They should also induce similar amount of cognitive load.

**RQ2.2: Does each variation of FA have different interaction with the environments?** The slight modifications to the original FA technique may induce different interaction with the environment.

**RQ2.3: Does each variation of FA induce different target acquisition paths?** In particular, does the orbital +Arc encourage different acquisition paths than the more direct standard FA?

**RQ2.4: Does each variation of FA have different subjective impression?** Although FA seems to have a good subjective impression in Gruenefeld et al. [8], we may be able to observe differences in the variations.

### 6.2 Results

#### 6.2.1 Number of Unsuccessful Target Acquisition

To answer **RQ2.1–2.2**, we fitted a mixed multiple logistic regression model that models the probability of failing a trial using the participants as the random effect with `lme4`. We obtained the following model: $\mathrm{logit}(\mathrm{Pr}(Fail)) = -5.04 + 0.96 \times Arc + 0.9 \times Trail \times 0 \times None - 0.5 \times Mixed - 1.10 \times Arc : Trail + 0.22 \times Arc : None + 1.08 \times Arc : Mixed - 0.56 \times Trail : None + 0.21 \times Trail : Mixed + 0.26 \times Arc : Trail : None - 0.14 \times Arc : Trail : Mixed$. The tests for the coefficients are summarized in Table 3. The $r^2$ for the models computed with `MuMIn` were: theoretical marginal = 0.06, theoretical conditional = 0.43, delta marginal = 0.01, and delta conditional = 0.07. The most important $r^2$ is theoretical conditional $r^2$ which represents goodness of fit of all terms in the model including the random effect. In this case, the effect size was moderate.

The main effects +Arc ($Z = 2.22, p \le 0.03, OR = 2.61$), and +Trail ($Z = 2.06, p \le 0.05, OR = 2.46$) were statistically significant–

|  | β | OR | SE | Z | p |
|---|---|---|---|---|---|
| (Intercept) | -5.04 | 0.01 | 0.49 | -10.39 | 0.00* |
| **+Arc** | | | | | |
| TRUE | 0.96 | 2.61 | 0.43 | 2.22 | 0.03* |
| **+Trail** | | | | | |
| TRUE | 0.90 | 2.45 | 0.44 | 2.06 | 0.04* |
| **Environments** | | | | | |
| None | 0.00 | 1.00 | 0.51 | 0.01 | 0.99 |
| Mixed | -0.50 | 0.61 | 0.57 | -0.86 | 0.39 |
| **+Arc : +Trail** | | | | | |
| TRUE:TRUE | -1.10 | 0.33 | 0.56 | -1.95 | 0.05* |
| **+Arc : Env.** | | | | | |
| TRUE:None | 0.22 | 1.24 | 0.60 | 0.36 | 0.72 |
| TRUE:Mixed | 1.08 | 2.94 | 0.65 | 1.65 | 0.10 |
| **+Trail : Env.** | | | | | |
| TRUE:None | -0.56 | 0.57 | 0.64 | -0.87 | 0.39 |
| TRUE:Mixed | 0.21 | 1.24 | 0.68 | 0.31 | 0.76 |
| **+Arc : Trail : Env.** | | | | | |
| TRUE:TRUE:None | 0.26 | 1.30 | 0.81 | 0.32 | 0.75 |
| TRUE:TRUE:Mixed | -0.14 | 0.87 | 0.82 | -0.17 | 0.87 |

Table 3: The coefficients, their associated ORs, and tests for the mixed multiple logistic regression predicting the probability of failing a trial. * signifies that $p \leq 0.05$.

|  | β | SE | df | t | p |
|---|---|---|---|---|---|
| (Intercept) | 2.17 | 0.09 | 40.11 | 23.16 | 0.00* |
| **+Arc** | | | | | |
| TRUE | 0.38 | 0.04 | 5527.98 | 8.70 | 0.00* |
| **+Trail** | | | | | |
| TRUE | -0.12 | 0.04 | 5527.99 | -2.89 | 0.00* |
| **Environments** | | | | | |
| None | -0.08 | 0.04 | 5527.99 | -1.76 | 0.08 |
| Mixed | -0.02 | 0.04 | 5527.98 | -0.50 | 0.62 |
| **Angular Distance** | | | | | |
| Dist | 0.01 | 0.00 | 5530.64 | 18.52 | 0.00* |
| **+Arc : +Trail** | | | | | |
| TRUE : TRUE | 0.17 | 0.06 | 5527.98 | 2.72 | 0.01* |
| **+Arc : Env.** | | | | | |
| TRUE:None | 0.15 | 0.06 | 5527.98 | 2.42 | 0.02* |
| TRUE:Mixed | -0.01 | 0.06 | 5527.98 | -0.20 | 0.84 |
| **+Trail : Env.** | | | | | |
| TRUE:None | 0.03 | 0.06 | 5527.98 | 0.45 | 0.65 |
| TRUE:Mixed | 0.02 | 0.06 | 5527.98 | 0.30 | 0.77 |
| **+Arc : +Trail : Env.** | | | | | |
| TRUE:TRUE:None | -0.25 | 0.09 | 5527.98 | -2.92 | 0.00* |
| TRUE:TRUE:Mixed | -0.03 | 0.09 | 5527.99 | -0.33 | 0.74 |

Table 4: The summary of coefficients and the results of the tests for the coefficients for the time required to reach out-of-view targets. We only considered successful trials for analysis. * signifies $p \leq 0.05$.

meaning that +Arc and +Trail increased the probability of missing the target. Meanwhile, the interaction between +Arc and +Trail was borderline statistically significant ($Z = -1.95, p = 0.051$). This meant that a combination of +Arc and +Trail was better than a variation with just one of the two behaviours. The environment did not affect performance for any variation of FA–meaning that despite the different cue trajectories and visual effects, the variations were relatively resistant to the visual complexities the environments.

### 6.2.2 Time for Target Acquisition

To answer **RQ2.1–2.2**, we fitted a mixed multiple linear regression model that predicted the time for successful target selection using `lme4`. We fitted the following model: $Time = 2.17 + 0.38 \times Arc - 0.12 \times Trail - 0.08 \times None - 0.02 \times Mixed + 0.01 \times Dist + 0.17 \times Arc : Trail + 0.15 \times Arc : None - 0.01 \times Arc : Mixed + 0.03 \times Trail : None + 0.02 \times Trail : Mixed - 0.25 \times Arc : Trail : None - 0.03 \times Arc : Trail : Mixed$. Table 4 shows the results of the tests on the coefficients. The pseudo-$r^2$ that we computed using `MuMIn` were as following: marginal = 0.12, conditional = 0.36. The conditional $r^2$ was considered moderate. While distance is statistically significant ($\beta = 0.01, t(5530.64) = 18.52, p \leq 0.05$), it did not contribute much in term of time required to reach the targets. We found that in general, +Trail increased the speed of target while +Arc slowed down the participants. To better explain how +Trail and +Arc affected the speed, we created a supplementary heatmap (Figure 8) that represents the average speed of target acquisition per condition.

### 6.2.3 Straightness

To answer **RQ2.3**, we performed a three-way repeated measure ANOVA (using `lme4`) on normalized $d$ of successful trials. We used `bestNormalize` for the normalization process. The factors were: +Arc, +Trail, and the environments. The significant effects were: +Arc ($F(1, 5529.2) = 81.58, p \leq 0.05$) and +Trail ($F(1, 5529.0) = 13.55, p \leq 0.05$). The test for the environments ($F(1, 5529.0) = 1.74$) was not statistically significant as well as the tests for interactions (+Arc:Trail – $F(1, 5529.0) = 0.32$, +Arc:Environment – $F(2, 5528.9) = 0.74$, +Trail:Environment – $F(2, 5528.9) = 1.02$, +Arc:+Trail:Environment – $F(2, 5528.8) = 1.32$). The subsequent post-hoc tests with `emmeans` on +Arc ($c = 0.23, Z_{ratio} = 9.03, p \leq 0.05$) and +Trail ($c = -0.10, Z_{ratio} = -3.68, p \leq 0.05$) were statistically significant. Figure 9 represents an interaction plot of the results.

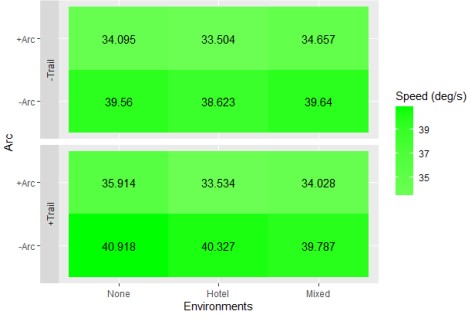

Figure 8: The average speed of target acquisition per condition. The unit is degree per second. Darker green means faster target acquisition.

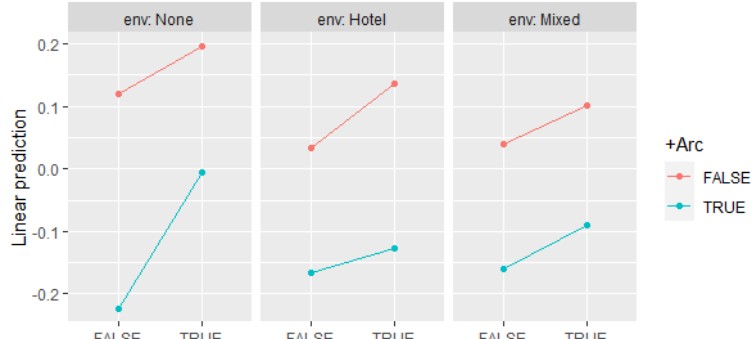

Figure 9: An interaction plot representing the tests of normalized $d$ in the second part of the study. env = Environment. As the y-axis representing normalized $d$, the plot does not represent descriptive statistics. Rather, this chart is to aid in interpretation of ANOVA results in Section 6.2.3.

We observed that while +Arc made target acquisition trajectories more circuitous while +Trail made them more direct.

### 6.2.4 Cognitive Load

To further answer **RQ2.1**, we analyzed the cognitive load collected using NASA-TLX questionnaire. After each participant completed a technique with all environments, we collected their raw NASA TLX-Score. The median scores were as following–FA-Arc-Trail: 54, FA-Arc+Trail: 57, FA+Arc-Trail: 48, FA+Arc+Trail: 57. Using repeated measure ART ANOVA with `art`, we found that the interaction between +Arc and +Trail was not statistically significant ($F(1, 69) = 0.82$), and neither was the main effects: +Arc ($F(1, 69) = 0.11$) and +Trail ($F(1, 69) = 0.92$). Despite these results, we could not conclude that the techniques induced roughly the same amount of cognitive load either, because we observed that the median for FA+Arc-Trail was much lower than the ones for the other variations.

### 6.2.5 Questionnaire

Overall, apart from some statements (For example, S10), the participants did not really indicate much different. Observing Figure 10, we note that the distributions of the scores tend to centre around higher numbers or somewhat uniformly distributed. Therefore, the questionnaire results did not provide a clear answer to **RQ2.4** except in a few cases.

**S1:** On average, participants indicated that all variations were almost as effective as each other (*Mdn*: FA-Arc-Trail = 7, FA-Arc+Trail = 7, FA+Arc-Trail = 6, FA+Arc+Trail = 6).

**S2:** On average, participants indicated that all variations were as understandable as each other (*Mdn*: FA-Arc-Trail = 7, FA-Arc+Trail = 7, FA+Arc-Trail = 6, FA+Arc+Trail = 6).

**S3:** On average, participants indicated that all techniques were as precise as each other (*Mdn*: FA-Arc-Trail = 7, FA-Arc+Trail = 7, FA+Arc-Trail = 6, FA+Arc+Trail = 6).

**S4:** The participants, on average, indicated that +Arc techniques were slightly more helpful (*Mdn*: FA+Arc-Trail = 5, FA+Arc+Trail

= 5) and the techniques that traveled in straight line were more effective (*Mdn*: FA-Arc-Trail = 7, FA-Arc+Trail = 6).

**S5:** The participants found it was the most difficult to concentrate with FA-Arc+Trail (*Mdn* = 4.5). The second worst technique is FA-Arc-Trail (*Mdn* = 3.5). The third worst technique is FA+Arc+Trail (*Mdn* = 3). The best technique is FA-Arc-Trail (*Mdn* = 2). The interview data indicated that the trail might have made it difficult to concentrate; five participants indicated so during the interview for FA-Arc+Trail, and three participants indicated so during the interview for FA+Arc+Trail. We believe that +Arc might make it somewhat easier to concentrate, because the cue never got close to the participants.

**S6:** FA-Arc+Trail was the most startling variation (*Mdn* = 5), closely followed FA-Arc-Trail (*Mdn* = 4) and FA+Arc-Trail (*Mdn* = 3). The least startling variations was FA-Arc+Trail (*Mdn* = 2).

**S7:** On average, participants indicated that all techniques were almost as effective as each other at getting their attnetion (*Mdn*: FA-Arc-Trail = 6, FA-Arc+Trail = 7, FA+Arc-Trail = 6.5, FA+Arc+Trail = 6).

**S8:** We found that +Trail techniques (*Mdn*: FA-Arc+Trail 6.5, FA+Arc+Trail = 5) to be more obstructive than their -Trail counterpart (*Mdn*: FA-Arc-Trail = 4.5, FA+Arc-Trail = 4).

**S9:** On average, participants indicated that all techniques were almost as effective as each other at making them aware of objects outside the FoV (*Mdn*: FA-Arc-Trail = 6, FA-Arc+Trail = 7, FA+Arc-Trail = 6, FA+Arc+Trail = 6).

**S10:** On average, participants indicated that all FA+Arc+Trail to be the most comfortable (*Mdn* = 2) followed with FA-Arc-Trail (*Mdn* = 2.5). The third best technique was FA+Arc-Trail (*Mdn* = 3) and the worst technique was FA-Arc+Trail (*Mdn* = 4). We think the arc trajectory made the trail more comfortable to use, because the trail was never close to the participants.

## 7 DISCUSSION

### 7.1 Experiment 1

In Experiment 1, the results indicated that bSOUS is a viable technique. bSOUS is faster and less obstructive than FA. This also highlights the benefit of a technique with an interface solely inside the user's peripheral vision. Since the cue will always be inside the user's peripheral vision, we do not need to worry about positioning the cue like FA or 3D Wedge. Although the speed increment by SOUS is small, we still think it is still important for certain scenarios such as competitive gaming where any increment is important.

We found the effectiveness of fSOUS is very sensitive to the visual complexity of the environment. Although fSOUS can help the user to locate targets quickly in a less visually complex environment, it can end up hindering the user in more complex environments. This contradicts other works that have successfully used faint cues to guide the user such as Rothe and Hußmann [22, 23] and McNamara et al [16]. One notable difference between our work and these is that fSOUS is strictly inside the user's far-peripheral vision while the other techniques have cues that are closer to the user's macular vision. Our results are more in line with other findings (such as [15, 20]) which show discerning details in far-peripheral vision is difficult. Therefore, we conclude that while a faint technique can be effective, it tends to negatively interact with the visual scene and becomes less effective in the peripheral vision. We also recommend a faint cue to be used only in scenarios where it will not be restricted inside the user's peripheral vision. If the cue must be within the user's peripheral vision, then the scene must not visually complex. We also suggest using a faint cue in a low-stake task where successful target acquisition is not paramount to the task as a whole. For example, the user may just be exploring a virtual museum at their own pace and does not benefit from interacting with virtual museum pieces.

Gruenefeld et al. [8] argue that since FA has low usability (as measured by System Usability Score), it may explain why it is

| Q | +Arc | +Trail | 1 | 2 | 3 | 4 | 5 | 6 | 7 |
|---|---|---|---|---|---|---|---|---|---|
| 1 | - | - | 0 | 0 | 0 | 0 |  | 3 4 | 17 |
|  | - | + | 0 | 0 | 0 | 0 |  | 3 8 | 13 |
|  | + | - | 0 | 0 | 0 | 3 |  | 6 8 | 7 |
|  | + | + | 0 | 0 | 0 | 2 |  | 7 6 | 9 |
| 2 | - | - | 0 | 0 | 0 | 1 |  | 4 3 | 16 |
|  | - | + | 0 | 0 | 0 | 1 |  | 2 6 | 15 |
|  | + | - | 0 | 0 | 0 | 6 |  | 3 6 | 9 |
|  | + | + | 0 | 0 | 1 | 6 |  | 2 6 | 9 |
| 3 | - | - | 0 | 0 | 0 | 1 |  | 2 8 | 13 |
|  | - | + | 0 | 0 | 0 | 0 |  | 3 4 | 17 |
|  | + | - | 0 | 0 | 2 | 3 |  | 5 7 | 7 |
|  | + | + | 0 | 1 | 1 | 1 |  | 5 7 | 9 |
| 4 | - | - | 0 | 0 | 0 | 2 |  | 1 6 | 15 |
|  | - | + | 0 | 0 | 0 | 3 |  | 1 9 | 11 |
|  | + | - | 1 | 1 | 2 | 5 |  | 4 6 | 5 |
|  | + | + | 0 | 2 | 1 | 5 |  | 10 1 | 5 |
| 5 | - | - | 5 | 5 | 2 | 2 |  | 4 5 | 1 |
|  | - | + | 1 | 3 | 5 | 3 |  | 4 4 | 4 |
|  | + | - | 6 | 7 | 2 | 2 |  | 2 4 | 1 |
|  | + | + | 3 | 8 | 4 | 2 |  | 1 2 | 4 |

| Q | +Arc | +Trail | 1 | 2 | 3 | 4 | 5 | 6 | 7 |
|---|---|---|---|---|---|---|---|---|---|
| 6 | - | - | 2 | 7 | 2 | 5 | 6 | 1 | 1 |
|  | - | + | 0 | 3 | 0 | 4 | 8 | 4 | 5 |
|  | + | - | 6 | 8 | 1 | 4 | 3 | 1 | 1 |
|  | + | + | 4 | 7 | 2 | 3 | 4 | 2 | 2 |
| 7 | - | - | 1 | 1 | 0 | 4 | 2 | 5 | 11 |
|  | - | + | 0 | 0 | 0 | 1 | 1 | 9 | 13 |
|  | + | - | 0 | 0 | 1 | 3 | 2 | 6 | 12 |
|  | + | + | 0 | 0 | 0 | 0 | 3 | 10 | 11 |
| 8 | - | - | 3 | 1 | 5 | 3 | 7 | 1 | 4 |
|  | - | + | 0 | 0 | 0 | 0 | 1 | 11 | 12 |
|  | + | - | 5 | 5 | 1 | 5 | 3 | 2 | 3 |
|  | + | + | 1 | 5 | 1 | 4 | 3 | 7 | 3 |
| 9 | - | - | 1 | 0 | 0 | 0 | 3 | 9 | 11 |
|  | - | + | 0 | 0 | 0 | 0 | 2 | 8 | 14 |
|  | + | - | 0 | 1 | 0 | 1 | 4 | 10 | 8 |
|  | + | + | 0 | 0 | 0 | 1 | 2 | 11 | 10 |
| 10 | - | - | 3 | 9 | 2 | 6 | 3 | 0 | 1 |
|  | - | + | 1 | 5 | 3 | 4 | 3 | 5 | 3 |
|  | + | - | 5 | 5 | 4 | 5 | 3 | 2 | 0 |
|  | + | + | 6 | 7 | 4 | 5 | 0 | 2 | 0 |

Figure 10: The heatmap represents the frequencies of the responses for the Likert scale statements. Red means less participants and green means more participants. The numbers indicate the frequencies of responses.

slower than EyeSee360, another technique in their study. However, our study shows that there might be an alternative explanation to the relatively lower speed of FA. Some participants in our study believed that FA gave them a specific timeframe to complete the task while some said that the cue influenced their speed of target acquisition. Therefore, we argue that FA may not be a technique for maximizing target acquisition speed, but to control it. To truly verify this though requires a future study.

## 7.2 Experiment 2

While +Arc made FA less obtrusive and thus more similar to SOUS, we did not observe an increase in speed like SOUS. Instead, it slowed down the participants even further. FA also did not become more comfortable to use despite less visual obstruction. However, we believe that we can increase comfort by making the cue adjusting its own trajectory to maximize comfort. For example, a cue takes a path around the user instead of above the user to reach the target. Interestingly, while the first study implementation of FA has a consistent behaviour through all environments, +Arc makes FA more sensitive to the visual complexity in the environment in term of speed–somewhat similarly to fSOUS. We require further investigation to find the reason behind the increased sensitivity.

+Trail improved the speed of target acquisition through increased persistence of the cue. However, we also found the participants to be less successful at acquiring out-of-view targets. The interview data revealed the participants did not find the trail comfortable to use. They suggest that the trail should be smaller, and more translucent so they can better see the surrounding. Based on bSOUS and fSOUS being faster than FA in the first study and the improvement brought by +Trail, we suggest that a more visible cue may reduce target acquisition time. A more visible cue also makes target acquisition trajectories more direct.

Overall, this study suggests that placing an interface completely inside a user's far-peripheral vision provides the best balance between obtrusiveness and visual persistence. However, if the user has to use a low-FoV HMD that is incapable of displaying beyond the user's mid-peripheral vision, something similar to +Trail may be useful to increase their speed of target acquisition. However, one must be aware that a +Trail technique requires fine-tuning of the cue to ensure an optimal experience.

## 7.3 What Is a Good Technique?

Upon analyzing our results, it appears that each technique could be useful in different contexts. However, as reflected in our research questions, we maintain that our set of desirable characteristics of off-screen target cuing hold in most cases: (1) make the user more successful at target acquisition while inducing low cognitive load at the "optimal" speed, (2) not be impacted by the visual complexity of a scene, and (3) provide a good subjective user experience. It is important to note that "optimal" speed does not always mean "fastest." Rather, this may be the speed that will lead to the best user experience. For example, in competitive gaming, the highest speed is likely better whereas in a VR museum exhibit, a slower speed might be desirable to allow the user to observe the environment along a trajectory. We think that bSOUS and fSOUS are appropriate for maximizing speed whereas FA may be viable for reducing and controlling the speed.

## 7.4 Limitations and Future Work

### 7.4.1 Realistic Tasks

While the results of the work suggested that each technique could be appropriate for certain task, our study tasks of acquiring in-view and out-of-view targets were generic. A future study could investigate the techniques along with more realistic scenarios, such as playing an on-rail shooter game that requires the participants to search for and destroy the targets as quickly as possible. Furthermore, to evaluate

FA's potential use as a tool to control the user's target acquisition speed, we could ask participants to acquire a target at a specific pace and observe how well they conform to our request with and without the technique.

### 7.4.2 Multiple Target Acquisition

Our study only involved single-target acquisitions. However, in more realistic scenarios, multiple targets may be involved. Therefore, in a future study, we could also explore adding more out-of-view targets per trial and observe how participants' attempts to reach and acquire the targets.

### 7.4.3 Eye Tracking

Currently, bSOUS and fSOUS ensure that the cue are outside the user's FoV solely based on the user's head orientation. However, the user's gaze can also saccade or move around which could bring their macular vision closer to the target. We originally planned to use Pupil [13], an eye tracking hardware add-on, to keep track of the user's gaze to make additional adjustment based on the gaze position. However, the device is not compatible with Pimax 5K Plus, the HMD. While we possess HTC Vives, devices that are compatible with Pupil, their FoV is insufficiently wide for bSOUS and fSOUS. We hope that future advancement in HMD and eye tracking devices could allow us to incorporate eye tracking in designing and evaluating bSOUS and fSOUS.

## 8 Conclusion

We conducted a two-part study in which participants selected out-of-view targets with the aid of visual cuing techniques. In the first experiment, we found that bSOUS and fSOUS had a reasonable performance when compared to FA. However, fSOUS has a significant weakness: the cue tends to be not sufficiently salient against the environments. Overall, the first experiment demonstrates that a technique whose interface is completely inside the user's far-peripheral vision can be effective. Although SOUS has a simple and a straightforward design, far-peripheral techniques like SOUS were impossible to evaluate until recently due to limited FoV of the commodity HMDs. In the second experiment, we modified FA to make cue trajectories travel relative to the user (+Arc) and make it more noticeable (+Trail), so that FA can be less obtrusive and more persistent like SOUS. Overall, +Arc decreased the effectiveness of FA while +Trail increased the speed of the participants while reducing the chance of acquiring the target. This means that decreasing obtrusiveness does not necessarily lead to a desirable behaviour and increasing persistence can reduce target acquisition time. We suggest that a technique that exclusively uses the user's far-peripheral vision has the best balance between obtrusiveness and persistence. However, if a HMD cannot effectively display beyond the user's mid-peripheral vision, something akin to +Trail may be useful. Our study shows that there is no one-size-fit-all technique. When designing a technique or modifying an existing technique, we must consider multiple competing factors and scenarios.

### Acknowledgments

The authors wish to thank Chinenye Ndulue and members of Gem-Lab at Dalhousie University for their input while developing the software for the study. They also would like to thank the participants of the study. This work is funded by Natural Sciences and Engineering Research Council of Canada (NSERC).

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
