# OpenReview forum: "A Comparative Evaluation of Techniques for Locating Out of View Targets in Virtual Reality"
_graphicsinterface.org/Graphics_Interface/2021/Conference — GI 2021_

### Official Review · AnonReviewer1 · 2021-01-02
**Good comparisons on out-of-view VR target acquisition**

**Rating:** 6
**Confidence:** 3

**Review:**

The paper describes and analyzes comparisons of different techniques to indicate and locate out-of-view targets in VR environment. The authors generally prefer the two SOUS methods over the prior work, FA method, but they didn't claim contribution to SOUS. So the contribution of the paper is mostly the comparative study.

Overall the paper is well written, with details of the experiments recorded and analyzed. I think the technical contribution is weak, but the results may have good practical use.

Note: Figure 3 may be wrong. b should be the technique with trail c should be the one without trail.

---

### Official Review · AnonReviewer2 · 2021-01-11
**Techniques to guide user to out-of-view targets**

**Rating:** 5
**Confidence:** 4

**Review:**

The paper describes two experiments that evaluate different techniques to guide users towards a target outside their field of view. In the first experiment, they evaluate two techniques fSOUS (low visual salience) and bSOUS (visually salient) to guide users towards a target outside their field of view. They compare this technique against FlyingArrow, a previously proposed technique that uses an arrow inside the VE to guide the users. Results show their techniques to improve user performance over FlyingArrow in various environment types. In the second experiment, they test different modifications of FlyingArrow to improve their performance. They found that adding trails helped performance but were more intrusive.

This is a good paper, but some missing elements make judging the results difficult. Here are some questions I had while reading the paper:

For both experiments

1) Why the in-view (faded) and out-of-view (sparked) targets had different selection feedback? Also, in Figure 6, the out-of-view target is red vs yellow for the in-view target. Is this difference part of the user study? Different selection feedback might have distracted users, but it was not mentioned when discussing the results.
2) What are the target and cursor sizes, and why were those selected? Even if they are not part of the evaluated parameters, the user performance, especially if the selection is gaze-based.
3) What device was used to track the eyes during the study for gaze-selection? Also, include device latency in the paper.
4) Was the saccadic eye-movement considered for the data analysis?
5) Explain what a short break means in study 2 and include if study 1 participants also took a break.

For experiment 1

1) In section 5.6.2 the paper mentions that “We did not have to consider Fitts’s Law for this study, because our targets have the same angular size”. Yet, in the next paragraph, the paper discusses that the increase in angular distances did not follow Fitts’ Law. I suggest removing this sentence as the experiment did not follow the correct protocol to make this conclusion.
2) Some conclusions are too strong and not supported by the data. I recommend adding what data support these conclusions or removing the sentence. For example, "we argue that the true strength of FA is not about maximizing speed of target acquisition, but to limit it".

For experiment 2

1) Why was the modified FlyingArrow white and not red like Gruenefeld et al’ s paper? Colour has a large effect on visual cues, so I suspect this might affect the results, but it is not mentioned in the discussion.

Minor comments

1) It is difficult to understand the differences between FlyingArrow modifications from Figure 3

---

### Official Review · AnonReviewer3 · 2021-01-12

**Rating:** 6
**Confidence:** 4

**Review:**

This paper presents two AR techniques for out-of-view target guiding awareness and variants in two experiments with human participants. In the first experiment, two variants of the SOUS technique, bSOUS and fSOUS, were compared using a target selection task. Both techniques are built on the existing FA technique and are less intrusive than the original technique. The first experiment showed that they both perform better (although by a small margin) than FA. Furthermore, they found that bSOUS was more robust in complex environments than fSOUS. In the second experiment, the authors compared four variants of the FA technique in a second experiment in which two additional behaviors were added to FA. The experiment showed that each technique offered trade-offs in speed and accuracy and there was some indication that the complexity of the environment was also a factor in how effective a particular techniques is.

The paper is clearly written and the techniques and rationale for selecting them are well described. The quantitative data is well presented and future research can explore how these techniques perform in realistic applications. This brings to my concerns about the paper: I had a hard time determining if the paper is making a large enough contribution to the field to warrant publication at GI. The research, as presented, has not resulted in large gains (although some statistically significant results were observed) and it remains to be seen whether each technique's particular gains would impact the user experience in concrete applications (e.g., gaming, etc.). Having said that, I appreciate that the authors discuss some of these aspects in the paper and especially in Section 7.3.  I suggest the authors add a limitations section and provide some ideas for how the research can be extended in the future. Another issue that needs clarification is the relationship between experiment 1 and 2: Upon my first reading, I got the impression that experiment 1 informed experiment 2 but it seems like both experiments were completed at the same time and by the same participants. I recommend the authors revise the methods section to clarify this point further.

---

### Meta-Review · Area_Chair1 · 2021-01-15

**Recommendation:** Accept
**Confidence:** 3

**Metareview:**

Overall, the reviewers had positive responses to the paper, although they also had concerns about the technical contribution of the paper and have asked the authors to provide more details about different aspects of their study (including the relationship between the two studies. I will make an overall recommendation that the paper is considered a marginal accept and request that the authors pay close attention to the detailed feedback provided by the reviewers and incorporate them into the paper before final submission.

---

### Decision · Program_Chairs · 2021-01-16

Accept